# The Impact of Living Arrangements and Social Capital on the Well-Being of the Elderly

**DOI:** 10.3390/healthcare11142050

**Published:** 2023-07-17

**Authors:** Chun-Chang Lee, Ruo-Yu Huang, Yun-Ling Wu, Wen-Chih Yeh, Hung-Chung Chang

**Affiliations:** 1Department of Real Estate Management, National Pingtung University, Pingtung City 912301, Taiwan; 2Department of Real Estate Management, HungKuo Delin University of Technology, New Taipei City 236354, Taiwan; 3Department of Business Administration, Chihlee University of Technology, New Taipei City 220305, Taiwan

**Keywords:** living arrangements, social capital, ordered logit regression model, aging society, subjective well-being

## Abstract

This study examines the impact of living arrangements and social capital on the subjective well-being of the elderly, as well as the mutual effects and relationships between the well-being and self-rated health status of the elderly. A total of 369 questionnaires were administered, and the effective recovery rate was 98.10%. The results indicate three key findings: (1) the current location for aging in place, social support, social activities, house ownership, and self-rated health status are indispensable factors affecting the well-being of the elderly. The best location for aging in place was the community, where the elderly’s sense of well-being was highest—the next best options were aging at home and institutional care. (2) Elderly people with sole ownership of their homes were more likely to have higher levels of well-being than those owning jointly or who were tenants. (3) There was significant interaction between subjective well-being and self-rated health status.

## 1. Introduction

In 2018, Taiwan became an aging society, as marked by the rapid aging of its population structure (Source: Indicators of a Super-Aged Society. Population Projections for the R.O.C. Website: ndc.gov.tw. Last accessed in 21 August 2020). In 2020, it became a super-aged society, with the super-aged—those aged 85 and above—accounting for 10.3% of the elderly population. The implicit significance of this phenomenon is that Taiwanese elderly people are making changes in their choice of living arrangements. Thus, the issue of whether an older person’s choice of living arrangements provides them with a healthy, comfortable, and satisfying space has become a universal issue of concern in Taiwan.

In the past, a vast body of research has examined the well-being of the elderly through personal attributes. Rodriguez-Blazquez et al. [1] suggested that age had a greater impact on the well-being of community-dwelling elderly than on the well-being of those who are institutionalized. Mackenzie et al. [2] examined the assumption that moral support is conducive to a sense of coherence and mental health and found that many people expressed that their relationship with God was the foundation of their mental health. Olivos [3] established that health is the most crucial determinant of well-being, followed by income and lifestyle. Lee et al. [4] examined the well-being and health-promoting lifestyle of elderly people living alone in Datong District, Taipei City. The participants were 259 elderly people aged 65 years and above and under the jurisdiction of the Department of Social Welfare. The results showed that those who were satisfied with their financial status had a higher level of well-being. In terms of health-promoting lifestyle, better self-realization, health responsibility, and interpersonal support positively influenced well-being.

In addition to personal attributes, social capital is also a salient factor affecting the well-being of the elderly. Social capital includes web-based interpersonal networks and assets available for use [5,6]. In other words, networks represent various social connections that link people and organizations. Helliwell and Putnam [7] pointed out that social capital is mainly derived from the strength of one’s relationship with family, neighbors, religion, and the community. Social capital also supports subjective well-being and physical health. Nyqvist et al. [8] consolidated and analyzed various studies on the effect of social capital on the mental health of those over 50. All the included studies demonstrated a positive correlation between social capital and mental health. Theurer et al. [9] examined altruistic behavior and social capital as predictors of well-being and life satisfaction among Canadians. The results indicated a strong association between social capital and well-being. Yeh et al. [10] explored the social support system and life satisfaction among elderly Kaohsiung City residents by 4894 representative and valid samples of the Supply and Demand for Elderly Health and Welfare Services in Kaohsiung City Survey developed by the Department of Health, Kaohsiung City Government in 2000. The results revealed that social support systems (social capital) significantly influence life satisfaction, as well-being increases with life satisfaction. Liang et al. [11] studied the influence of social capital on the quality of healthy living among 212 elderly residents living in their homes and attending Category C care centers in Taichung City. The results showed that the social capital of elderly people significantly influenced their quality of healthy living. The authors recommended that the government should include measures on increasing the social capital of elderly groups during policymaking.

A further axiomatic factor affecting the well-being of the elderly is their living arrangements. Aging in place (or AIP, i.e., aging in one’s own home and community) and institutional care affect the mental health and well-being of the elderly [12]. Previous studies on elderly people’s choice of living arrangements have shown that living with their children is the ideal arrangement. Huang and Hsia [13] demonstrated the important consequences of the close relationships between elderly people’s family of origin and their decision to live in the same house or community as their children.

This study examines three types of living arrangements—aging in place (aging in one’s own house and aging in the community) and institutional care—in terms of how these affect the well-being of the elderly. In general, well-being is associated with interpersonal relations and social capital. We concurrently explored the effects of living arrangements and social capital (social support and social activities) on the well-being of the elderly. Based on the literature, there are also endogenous effects resulting from relationships between variables that are not simply one-way causal relationships. Some studies have demonstrated that health affects well-being (see [3,14,15]), while others have argued that well-being affects health. In the case of the latter, Chiang and Lee [16] showed that well-being is an important predictor of perceived health. Indeed, the relationship between health status and the well-being of the elderly is not one-way. The well-being of the elderly affects their mental health, and the latter is thus not an exogenous variable. In our empirical model, we consider the possible relationship between well-being and health status. Using a multinational sample of 49,504 people across 30 countries, Wang and Wend [17] examined the influence of social capital on well-being in different age groups. The results demonstrated that for elderly people (those above 65 years of age), a significant interaction existed between the degree of urbanization and well-being. This suggests that countries with more social capital can reduce the influence of urbanization on well-being. Furthermore, Chen and Lin [18] surveyed the relationship between living arrangements and life satisfaction in Taiwanese elderly people. The results showed that soaring property prices may result in regional differences in life satisfaction among the elderly. However, their choice of living arrangements should be respected, and the government should provide suitable special assistance to economically disadvantaged elderly groups to improve their life satisfaction. The literature review suggests that living arrangements and social capital profoundly influence the well-being of the elderly, especially in regional living environments. Therefore, the uniqueness of this study is its inclusion of endogenous problems such as the mutual effects of the well-being and self-rated health status of elderly people, as well as the analysis of their personal attributes, house ownership, living arrangements, and social capital based on the housing characteristics of ethnic Chinese people. We also used an ordered logit regression model analysis to analyze the subjective well-being of the elderly because this approach is frequently used in gerontology studies, such as research on older adult volunteering and active aging [19] and the influence of social capital and real estate property ownership on the well-being of elderly people [20].

## 2. Literature Review

### 2.1. Personal Attributes

Considerable research has shown that personal attributes (including marital status, religion, financial status, health status, and education level) are associated with well-being. Jeon et al. [21] reported that strong family ties, financial stability, and health are the most important factors affecting the well-being of community-dwelling elderly. Elderly people with poorer health have lower life satisfaction [10]. Huang and Yang [22] reported that elderly people are more capable of attaining a sense of well-being when they are well-educated, financially well-off, and healthy and have many sincere friendships. Kehn [23] showed that the explanatory variables of health, marital status, and religion were significantly correlated with the well-being of the elderly and are suitable predictors. In particular, regarding health status, some results support that well-being affects health status, such as the study by Chiang et al. [16], in which well-being was an important predictor of self-rated health. Other studies argue that health affects well-being, such as Kim et al. [14], in which the authors studied the predictors of well-being in older Korean women and showed that well-being was associated with mental and physical health. Angner et al. [15] concluded that health status was one of the most salient predictors of subjective well-being. Juang et al. [24] studied the importance of the influence of personal attributes and different personal backgrounds of elderly groups (those aged 65 years and above). The results showed that personal background, gender, place of residence, age group, education level, and marital status significantly influenced the well-being of different elderly groups. To summarize, health status, education, and financial status are important factors influencing the well-being of elderly people. We thus suggest that variables such as marital status, religion, financial status, health status, and education level can better highlight the influence of personal attributes on well-being.

### 2.2. Social Capital

Social capital is defined as the empathy or obligations expressed by a person or group toward another person or group. Cojan [25] noted that social capital theory stresses social participation and trust, mutual assistance, and communication between neighborhoods, as these factors promote social cohesion. Li et al. [26] revealed that social capital is measured based on organizational participation, trust, reciprocity, and neighborhood relations. Previous studies have reported that emotional support affects the well-being of the elderly. Heller and Swindle [27] suggested that social support is drawn from society (school, church, etc.), personal networks (friends, classmates, etc.), and significant others. Graham et al. [28] showed that villages offered a promising new model of support and positive effects to community residents, which may reduce loneliness, promote well-being, and enhance the confidence of those aging in place. Jheng [29] indicated that interpersonal networks, social capital, and social connections are positively and significantly correlated with psychological well-being. Hwang and Sim [30] evaluated participation in social activities, including religious, leisure, and charitable activities and social belongingness. The findings indicated that those who regularly participated in various activities were more likely to experience greater well-being. Wu et al. [31] studied how the self-rated health of elderly people living in the community in Taiwan influenced their quality of life through social capital. The results showed that their quality of life was associated with age, gender, education level, financial status, kinship, living environment, opportunities, and satisfaction. The influence of health status on quality of life is mediated by social capital. Social capital is an important determinant of the well-being of elderly people. In short, social capital is mainly rooted in society and personal networks and established on mutual assistance and trust between people. Social capital positively influences the well-being of elderly people.

### 2.3. Living Arrangements

Regarding living arrangements, Yen and Anderson [32] have defined “aging in place” as an elderly person living in their own home or community in the long term. The “place” in the phrase primarily refers to a place from which elderly people are not displaced; they remain in their existing residence and age while enjoying their established lifestyle. There are two aspects of aging in place: most directly, aging in one’s home and then aging in the community shaped by the social circle around that home.

Golant [33] critiqued the concept of aging in place, suggesting that a single aging-in-place solution may not be optimal for low-income and frail elderly Americans. The willingness of elderly Taiwanese to age in place is evidenced in statistics published by the Ministry of Health and Welfare’s 2017 Survey on Elderly Groups’ Status; these show that only 12.70% of elderly interviewees over 65 who were capable of self-care were willing to live in a nursing facility or a senior-living apartment or housing. Grimmer, Kay, Foot, and Pastakia [34] showed that community-dwelling elderly in Australia were more optimistic about aging in their own homes as long as they received what they actually needed instead of what people assumed they needed.

Next, regarding studies on institutional care, Zhang [35] explored the influence of living arrangements (living alone, with family members, in an institution) and found that those who lived alone or in an institution had negative well-being. Wu [36] pointed out that elderly Nanjing residents living in institutions generally experienced better well-being, despite the differences between institutions. Hsiao [37] showed that elderly Taiwanese living in welfare institutions in Pingtung Country had higher well-being when they had better social support. The quantitative study by Chen [38] found that daycare services indeed affect the well-being of the elderly. Although aging in place has become a mainstream option worldwide, it might not be the best elder-care solution. Lai [39] evidenced that the number of family caregivers is gradually dwindling as society rapidly ages; for many families, institutional care is the only living-arrangement option for their elderly members. While institutional care has the advantages that come with long-term elder care, recipients must be displaced from their original dwelling to a long-term care center or nursing facility. Kuo [40] studied the impacts of high housing prices on intergenerational living arrangements, using the follow-up data provided by a dynamic database of ethnic Chinese families. The results showed that rising housing prices in urban areas can drive married children to eventually move out of their parents’ homes. Additionally, determinants of co-residence were as follows: the age of the married child; whether the child is the only child of the firstborn; the presence of preschool-aged children in the family; a deceased parent; residential status. To summarize, living arrangements (living alone, living with family, or living in a care facility) is an important determinant of well-being for elderly people.

## 3. Methods

### 3.1. Empirical Model Settings

This study examines whether various factors affect the subjective well-being (SWB) of the elderly by applying an ordered logit regression model for analysis. Well-being in this study was measured on an ordinal scale with three options: disagree, neutral, and agree. Assuming that well-being is represented by $Y^{*}$ (Y=1 indicates disagreement; Y=2 indicates neutrality; Y=3 indicates agreement), and x is an aggregate vector of the independent variables (sex, age, education level, family type, household income, health status, religion, house ownership, area, location, living arrangements, social support, and social activities), the regression model of the latent variable $Y^{*}$ is then expressed as follows (see [41]):(1)$$Y^{*}=\beta^{\prime }x+\varepsilon$$
where ε is an error term assumed to be normally distributed, and $Y^{*}$ cannot be directly explained through an independent variable; it may be confounded by other variables omitted from the above list but whose order can be determined. Thus, the model can further be expressed as follows:(2)$$\mathrm{Prob}\;\left(Y=3\right)=F\left(\beta^{\prime }x\right)\;\;\;\;\mathrm{if}\mu <\beta^{\prime }\;\mathrm{Prob}\;\left(Y=2\right)=F\left(\beta^{\prime }x+c\right)-F\left(\beta^{\prime }x\right)\;\;\;\;\mathrm{if}\beta^{\prime }x<\mu <\beta^{\prime }x+c\;\mathrm{Prob}\;(Y=1)=F(\beta^{\prime }x+c)\;\;\;\;\mathrm{if}\mu >\beta^{\prime }x+c$$
where c>0 is a parameter, and F represents the function of the cumulative probability distribution. The maximum likelihood method can be used to estimate the vector $\beta^{\prime }$ of the coefficient of the independent variable x.

The dependent variable in this study is well-being, represented by SWB in $Y^{*}$ in Equation (1). The factors affecting well-being include the individual’s self-rated health status (SRHS), sex (SEX), age (AGE), education level (EDU1, 2), marital status (MARRY), family type (FAMTYPE1, 2), household income (INCOME1, 2), religion, house ownership (COWNSHIP1, 2), location (LOCATION), living arrangements (LIVETYPE1, 2), social support (SUPPORT), and social activities (SACTION).

Health status is an endogenous variable because it is affected by well-being. Previous studies on the factors affecting health status have identified age, sex, education level, income, and marital status as explanatory variables [42,43,44,45]. In this study, the factors affecting health status include subjective well-being (SWB), sex (SEX), age (AGE), education level (EDU1, 2), marital status (MARRY), family type (FAMTYPE1, 2), income level (INCOME1, 2), exercise frequency (SPORT), smoking (SMOKE), alcohol consumption (ALCOHOL), betel nut chewing (BETEL) (it is currently known that betel nut chewing can cause oral submucous fibrosis, which is a precancerous condition; nitrosamine is created through the nitrosation of arecaidine and arecoline in the oral cavity, and these nitrosation derivatives are found to be tumor inducing in animal subjects; in Taiwan, betel nut chewing is the main cause of oral cavity, with an estimated 4500 people diagnosed with the disease annually and 2100 people dying from it [46] (Health Promotion Administration, Ministry of Health and Welfare, 2015, https://www.hpa.gov.tw/Pages/Detail.aspx?nodeid=1127&pid=1804. Last accessed: 18 March 2023)), medical facilities (MEDICAL), and living location (LOCATION). Since health (SRHS) affects well-being, and well-being also affects health, we have established a simultaneous model that is estimated in two stages.

### 3.2. Description of Variables

#### 3.2.1. Dependent Variable

In the literature on well-being (see [47,48]), the Oxford Happiness Questionnaire (QHQ), developed by Oxford University psychologists Hills and Argyle [47], has been widely adopted; it is regarded as the most effective instrument for measuring well-being. Well-being served as a dependent variable and was measured using the QHQ. The items were measured on a five-point Likert scale, with a higher score indicating better well-being. For the sake of data observation, the dimensionality was reduced, while the information was retained. Based on other studies using the five-point Likert scale for measurements (see [49,50]), the well-being scores were in the following ranges: 2.50 points or fewer (SWB = 1, low level of well-being), 2.51 to 3.50 points (SWB = 2, medium level of well-being), and 3.51 points and above (SWB = 3, high level of well-being).

Kim et al. [51] showed that SRHS is a predictor of future functional decline in the elderly. A large number of current studies focus on measuring the health status of the elderly based on their activities of daily living (ADLs) and instrumental activities of daily living (IADLs). In this study, health status was measured on a five-point Likert scale, with a higher score indicating better health. The SRHS scores were in the following ranges: 2.50 points or fewer (SRHS = 1, poor self-rated health), 2.51 to 3.50 points (SRHS = 2, fair self-rated health), and 3.51 points and above (SRHS = 3, good self-rated health).

Health status is also affected by the endogeneity of the impacts of well-being, and this can be treated using instrumental variables. We employed a two-stage regression in our estimations. In the first stage, the dependent variable of health status was estimated through the independent variables of subjective well-being, sex, age, education level, marital status, family type, income, exercise frequency, smoking/alcohol consumption/betel nut chewing habits, medical facilities, and location. In the second stage, the goodness of fit of health status in the first stage substituted health status in the well-being regression model.

#### 3.2.2. Independent Variables

Based on previous studies on the well-being of the elderly ([30,52,53], etc.), the influencing variables include the personal attributes of elderly people, their housing attributes, living arrangements (choice of aging in place location), and social capital (including social activities and social support). The variable settings are shown in Table 1.

(1)Personal attributes

With regard to the variable of sex, Pinquart and Sörensen [54] found that women tend to have lower life satisfaction, well-being, and self-esteem than men. After adjusting for the differences in age, Flores et al. [55] found that sex differences in the experience of well-being favored men and that these differences were mainly the result of elderly women having poorer mean health and financial status than elderly men. Kim [56] examined how sex affects the well-being of elderly people living alone and found that women had higher levels of well-being than men. In this study, sex was set as a dummy variable (men = 1, women = 0). The coefficient of the dummy variable of sex (SEX) is uncertain and could be either positive or negative.

With regard to the variable of age, Luchesi et al. [57] reported that the well-being of community-dwelling elderly in Sao Paulo State, Brazil, is affected by their age. Frijters and Beatton [58] examined the association between age and well-being, showing that the promoting effects of age on well-being were most significant at age 60, but they declined drastically after age 75. In this study, age (AGE) was set as a dummy variable (75 years and below = 1; 76 years and above = 0); the coefficient is expected to be positive.

The education level variable was considered by [52], who found that married or cohabiting ethnic Malays with a higher level of education may have a better sense of well-being. An et al. [59] interviewed 2345 healthy Taiwanese adults grouped by age as young, middle-aged, or elderly. The results indicated that those with a higher level of education level were more likely to experience a higher level of life satisfaction and well-being. Well-educated people are more capable of managing their life problems, but this advantage is more evident in middle-aged and elderly adults than in young adults. This is because young adults today have completed higher education. In this study, there were three education levels (high, medium, low) and two dummy variables (for EDU1, higher level education = 1, and 0 if otherwise; for EDU2, medium level education = 1, and 0 if otherwise), with a low level of education serving as the baseline for comparison. The coefficients of the dummy variables for levels of education (EDU1 and EDU2) are expected to be positive.

With regard to marital status, Tan et al. [52] showed that divorced or separated people have a significantly poorer level of well-being than married or cohabitating people. Yang and Leone’s [60] results support the argument that married people are happier than unmarried people. In this study, marital status was set as a dummy variable (married = 1, and 0 otherwise (separated/divorced/widowed)). The coefficient of the dummy variable of marital status (MARRY) is expected to be positive.

Family type is considered in [31], who analyzed the association between family type and the well-being of elderly Koreans for three family types: single, living with a spouse, and living with family. The results showed that the factors affecting well-being differed by family type, and elderly people living alone might have difficulty in obtaining satisfaction and well-being through their work. In this study, there were two dummy variables set for family type (for FAMTYPE1, living with spouse = 1, and 0 if otherwise; for FAMTYPE2, living with family = 1, and 0 if otherwise), and being single serving as the baseline case. The coefficients of the family-type dummy variables (FAMTYPE and FAMTYPE2) are expected to be positive.

With regard to the variable of personal monthly income, research has shown that it is a factor affecting the living satisfaction (well-being) and health status of the elderly. Cuong [61] indicated that a higher income promotes Vietnamese elderly groups’ satisfaction with life and other people. In this study, personal income was divided into three levels: low (less than TWD 22,000), middle (TWD 22,001 to TWD 38,000), and high (more than TWD 38,001) income, with low-income groups as the baseline for comparison. There were two dummy variables set for monthly income (for INCOME 1, high income = 1, and 0 if otherwise; for INCOME 2, middle income = 1, and 0 if otherwise). The coefficients of the dummy variables of monthly income (INCOME 1 and INCOME 2) are expected to be positive.

(2)Housing attributes

Housing attributes are explored by Zheng et al. [62], who concluded that owning a house affected the SWB of Chinese citizens positively. Similarly, D’Ambrosio [63] found that real estate value was a predictor of life satisfaction. In this study, real estate ownership was divided into sole ownership; joint ownership with a spouse or with parents, siblings, or children; and rental or spousal ownership, with the latter serving as the baseline for comparison. There were two dummy variables set (for COWNSHIP 1, sole ownership = 1, and 0 if otherwise; for COWNSHIP 2, joint ownership with a spouse or with parents, siblings, or children = 1, and 0 if otherwise). The coefficients of the dummy variables of real estate ownership are expected to be positive.

With regard to living location, Easterlin et al. [64] suggested that in countries with low levels of economic development, urban dwellers had significantly higher levels of life satisfaction than suburban dwellers, but this effect gradually diminished in developed countries; suburban dwellers’ life satisfaction is proximal or even exceeds that of urban dwellers. In this study, the living location (LOCATION) was set as a dummy variable (city center = 1, suburban areas = 0), with living in suburban areas set as the baseline for comparison; the coefficient of LOCATION is expected to be positive.

(3)Living arrangements

Current studies on the place of aging have highlighted the presence of trust and social connections in our daily social networks [65,66]. Community-based inclusive support and services enable every group to meet their aging-related needs [67]. Promoting elderly people’s sense of belonging in the community, including through participation in community activities (attending church, etc.), enhances their intention to stay within the community and feel better [53]. Helliwell and Putnam [7] determined that marriage and family, ties to friends and neighbors, workplace connections, and civic engagement (both individual and collective) were all associated with happiness and life satisfaction. Zhang [35] studied the living arrangements and SWB of elderly Chinese and found that, for elderly people, living alone or in an institution was strongly associated with negative well-being.

In this study, the choices of living arrangements for those who are aging include aging at home, aging in the community, and aging in a nursing facility, with living in a nursing facility as the baseline case. Two dummy variables were set (LIVETYPE 1: aging at home = 1, and 0 otherwise; LIVETYPE 2: aging in the community= 1, and 0 otherwise). The coefficient of the dummy variable of living arrangements (LIVETYPE) is expected to be positive.

(4)Social capital

With regard to social activities, Hwang and Sim [30] found that elderly people living with their spouses expressed an increase in well-being when they had more interactions with friends. The support received by elderly people from their family and friends and their life satisfaction were predictors of their well-being. In this study, the social activities in which elderly people participate include volunteering, leisure, learning, and religious activities. The mean social activity participation was scored on a scale of 1 to 4 for no participation, annual, monthly, and weekly participation, respectively. A score of 2.50 points and below indicates low participation, while a score of 2.51 to 4.00 indicates high participation, with low participation serving as the baseline for comparison. One dummy variable was set (high participation = 1, low participation = 0). The coefficient of the dummy variable of social participation (SACTION) is expected to be positive; i.e., a higher score for social participation predicts a higher level of well-being.

With regard to social support, Hwang and Sim [30] provide evidence that neighbors are a strong support system for elderly people who live alone or with their spouses, mainly because loneliness can be reduced through their interactions. Based on the social support questionnaires designed by [68,69], social support in this study includes support from a spouse and informational, emotional, and instrumental support; it is measured on a five-point Likert scale. The mean social support score was classified on a scale from 1 to 5. A score of 2.50 points and below indicates no contact, a score of 2.51 to 3.50 points indicates occasional contact, and a score of 3.51 points and above indicates frequent contact; the no-contact case serves as the baseline for comparison. Two dummy variables were set (for SUPPORT1, frequent contact = 1, and 0 if otherwise; for SUPPORT2, occasional contact = 1, and 0 if otherwise). The coefficients of the dummy variables of social support are expected to be positive; i.e., higher social support indicates a higher predicted level of well-being.

We included several independent variables in the SRHS model: SWB, sex, age, education level, marital status, family type, income, exercise frequency, smoking/alcohol consumption/betel nut chewing habits, medical facilities, and location. Matud et al. [70] found that an elderly individual’s sex plays a pivotal role in their mental health. Previous studies have examined the SRHS of the elderly and have found that the SRHS score decreases with age [71]. Lawrence et al. [72] showed that married people were healthier and lived longer than unmarried, divorced, or widowed people. Assari et al. [42] indicated that the likelihood of elderly people having good mental health is strongly correlated with having a high income.

Exercise frequency is assessed as occurring once weekly or less, and twice, thrice, four times, and five times or more weekly. The dummy variable SPORT was assigned the value of 1 for those who exercised thrice or more weekly, and 0 otherwise. We expect exercise frequency to have a significant and positive effect on SRHS. The accessibility of medical facilities, according to [73], has a significant positive effect on the health of the elderly, and this effect is more pronounced in the low-income elderly and those who live further away from medical facilities. In this study, accessibility of medical facilities (MEDICAL) was set as a dummy variable, which was set as 1 if these were present, and 0 otherwise. We expect the coefficient of accessibility of medical facilities to be positive. The variable settings are shown in Table 1.

### 3.3. Questionnaire Design

The questionnaire designed for this study included items concerning the participants’ well-being, health status, and social support, measured on a five-point Likert scale. The final section covered the participants’ basic information, such as sex, age, education level, marital status, personal monthly income, religion, social activities, and current living arrangements. The detailed questionnaire is available from the author upon request.

This study employs the 29-item OHQ to measure participants’ well-being. Hills and Argyle [47] extracted eight factors from the 29 items: life is rewarding, mentally alert, pleased with self, find beauty in things, satisfied with life, can organize time, look attractive, and have happy memories. After excluding the factor “mentally alert”, Shu [74] categorized the remaining seven factors into social adaptation status and psychological well-being, thus distinguishing between social and personal well-being. This study adopts the suggestions of [75] and revises the OHQ such that social well-being is understood as comprising interpersonal relations and life satisfaction. Personal well-being has four dimensions covering self-assurance and physical and mental status, with each dimension consisting of three items, along with an additional item on global well-being, for a total of 13 items.

With regard to SRHS, this study employed the Ministry of Health and Welfare’s Survey on Elderly Groups’ Status, in which elderly people’s health status was measured using 6 ADL items and 9 IADL items, and 1 item on SRHS, for a total of 16 items. The participants’ social support was measured using the questionnaires designed by [68,69]. Social support consists of four dimensions: support from spouse, informational support, emotional support, and instrumental support, with two items for each dimension, for a total of eight items. Based on the studies by [76,77,78,79], social activities consist of volunteer activities, leisure activities, learning activities, and religious activities, and the frequency of participation is either weekly, monthly, annual, or no participation.

## 4. Data Sources and Descriptive Statistics

The participants in this study were mainly elderly people above the age of 65 living in Pingtung City and Kaohsiung City. Taiwan has 22 administrative divisions, including 6 municipalities and 16 counties. Kaohsiung City is the third-largest metropolitan area among the six municipalities directly under the central government. Pingtung County is the second-largest metropolitan area among the 16 county–city administrative regions. The two administrative districts are geographically adjacent. Various life and administrative mutual assistance conditions are also closely integrated. In terms of the convenience of sampling and the consistency of sample characteristics, this is beneficial to the sampling operation of this study. Additionally, the total number of people over the age of 65 in the Kaohsiung metropolitan area is as high as 507,616, and the proportion of the elderly population in the city is as high as 18.56%, making it the second oldest city among the six metropolitan areas. Pingtung County generally has the same aging urban population. The total population of Pingtung County aged over 65 is as high as 156,805, accounting for 19.66% of the 16 counties and cities. It is the fourth most aging city among the 16 counties and cities. The ratio of the elderly aged over 65 in the two regions is higher than the national average of 17.81. The issue of the well-being of the elderly is thus an important issue of regional governance in these two regions.

Participants were recruited from community-based senior activity centers (such as Hechun Culture and Education Foundation and Tianliao Senior Center), community colleges, care centers, parks, and institutions. We administered the questionnaire in person, and we accompanied and assisted participants in completing the questionnaire to improve the recovery rate and reliability, avert difficulties in completing the questionnaire, and prevent delays in mailing in the completed surveys. The sample size must be considered during sampling because it affects the accuracy of the estimation results. This study assumed a tolerable error of 0.05 and a level of significance (α) of 10%. This means that under a 90% confidence level, the required sample size was 271, which was met because there were 362 valid questionnaires. The questionnaire survey was undertaken from 1 October 2021 to 30 November 2021, and a total of 369 questionnaires were administered; after removing invalid responses, there were 362 valid responses, indicating a 98.10% effective recovery rate. The representativeness of a sample entails non-response bias, which means that under any of the following circumstances, a researcher may not be able to obtain sufficient information from the sample, or the lack of specific types of representative samples in the questionnaire may affect the structural integrity of the sample and result in statistical bias. Armstrong and Overton [80] proposed the non-response bias test process to examine the presence of non-response bias in a sample. The recovered questionnaires were divided into two groups. The chi-square test of homogeneity [80] was used to check whether the ratio of the participants’ responses to their demographic data (sex, age, marital status, etc.) was homogenous or consistent. No differences existed between the two groups based on the test results, which means that the distribution of the sample sufficiently reflected the distribution of the population.

Regarding the participants’ personal attributes, the minimum age was 65 years, the maximum age was 95 years, and the mean age was 73.70 years. Of the final sample, 52.21% were male, and 47.79% were female; 28.18% had completed an elementary school education; 17.68% had received no higher than an elementary school education; 17.96% had received a junior high school education; and 16.85% had received a high school (vocational) education; while college graduates and postgraduates accounted for only 3.04%. Most of the participants were married (63.26%), followed by widows/widowers (29.28%), those who were separated or divorced (5.52%), and then unmarried people (5.52%). A majority had a low level of income (less than TWD 22,000), accounting for 54.14% of the sample, followed by middle-income groups (TWD 22,001 to TWD 38,000) at 27.90%, and then those with a high income (more than TWD 38,001) at 16.57%. Regarding bad habits, 10.29% were smokers, 8.44% were drinkers, 0.53% were betel nut chewers, and the remaining 80.74% had no such habits. Regarding exercise frequency, 34.81% exercised five times or more weekly, 20.72% exercised twice weekly, 17.68% exercised thrice weekly, 13.81% exercised four times weekly, and 12.98% exercised once weekly or less.

Regarding housing attributes, 40.61% had sole ownership of their house, 18.51% were tenants, 17.96% lived in homes fully owned by their spouses, 3.87% had joint ownership with their spouse, and 17.68% had joint ownership with a family member (not including their spouse). Regarding current living arrangements, 213 (58.84%) were aging at home, 95 (26.24%) were aging in the community, and 54 (14.92%) were receiving institutional care (please refer to Table 2).

## 5. Empirical Results

This study analyzes the well-being of the elderly through ordered logit regression modeling. The results of the test of parallel lines showed that, for well-being, $\chi^{2}$ = 21.192, *p* = 0.270, and significance > 0.05. The parallel lines assumption is supported, and the regression equations are parallel. The test for model fitness returned a value for $\chi^{2}$ = 271.577 (*p* < 0.001), which means that the regression coefficient of at least one independent variable is not 0. The test of parallel lines also indicated that for health status, $\chi^{2}$ = 22.167, *p* = 0.138, and significance > 0.05. The parallel lines assumption is supported, and the regression equations are parallel. The model fitting results showed that $\chi^{2}$ = 202.521 (*p* < 0.001), which means that the regression coefficient of at least one independent variable is not 0. The results are specified in Table 3.

The empirical results were compared with those of similar studies to highlight the similarities and differences, along with the uniqueness of this study. We subsequently analyzed the estimated well-being and self-rated health status.

### 5.1. Estimation of Well-Being

The results of the two-stage estimation are shown in Table 4. Regarding the participants’ personal attributes, the estimated coefficient of SEX was −0.204 and did not attain a level of significance. The estimated coefficient of AGE was −1.809 and did not attain a level of significance. A low education level served as the baseline for comparison, and the estimated coefficients of education level (EDU1 and EDU2) were −0.671 and 0.187, respectively; neither attained a level of significance. The estimated coefficient of marital status (MARRY) was 0.933 and attained a 5% level of significance.

The status of single served as the baseline case for family type (FAMTYPE). The estimated coefficients of family type (FAMTYPE1, for living with a spouse = 1, and 0 otherwise; FAMTYPE2, for living with family = 1, and 0 otherwise) were 0.048 and −0.129 for FAMTYPE1 and FAMTYPE2, respectively; neither attained a level of significance.

The estimated coefficients of personal income (INCOME 1, for high income = 1, and 0 otherwise; INCOME 2, for middle income = 1, and 0 otherwise) were 0.569 and 0.285 for INCOME 1 and INCOME 2, respectively; neither result was significant.

The estimated coefficient of SRHS was 1.502 and attained a 1% level of significance, indicating that good health had a positive effect on the well-being of the elderly.

Regarding the housing attribute of real estate ownership (OWNSHIPS), rental or spousal ownership was the baseline for comparison. The estimated coefficients of real estate ownership (COWNSHIP 1, for sole ownership = 1, and 0 otherwise; COWNSHIP 2, for joint ownership with spouse or with parents, siblings, and children = 1, and 0 otherwise) were 1.243 and 0.925 for COWNSHIP 1 and COWNSHIP 2, respectively; both attained a level of significance. The estimated coefficient of LOCATION was 0.117 but did not attain a level of significance.

Living in a nursing facility served as the baseline for comparison in the variable of living arrangements. The estimated coefficients of living arrangements (LIVETYPE 1, for aging at home = 1, and 0 otherwise; LIVETYPE 2, for aging in the community = 1, and 0 otherwise) were 1.155 and 1.955 for LIVETYPE 1 and LIVETYPE 2, respectively, and both attained a level of significance.

Regarding social capital, low social participation served as the baseline for comparison in the variable of social activity (SACTION). The estimated coefficient of SACTION was 1.507 and attained a 1% level of significance.

No contact served as the baseline for comparison in the variable of social support (SUPPORT). The estimated coefficients (for SUPPORT1, frequent contact = 1, and 0 otherwise; for SUPPORT2, occasional contact = 1, and 0 otherwise) were 1.393 and −0.082, respectively, and the former attained a 5% level of significance.

### 5.2. Estimation of Self-Rated Health Status

In the first phase, well-being was estimated with reference to personal habits (smoking, alcohol consumption, and betel nut chewing), weekly exercise frequency, access to medical facilities, participants’ basic attributes, housing attributes, and social capital. These results were then used to estimate the SRHS of the elderly alongside the participants’ basic attributes, housing attributes, and social capital. The estimation results for the second phase are shown in Table 3. The effects on SRHS of personal attributes, personal habits, exercise frequency, and housing attributes were not central to this study, and, for reasons of length, are not elaborated on. The estimated coefficient of SWB was 0.957 and attained a 1% level of significance. This shows that well-being had a significant and positive effect on the SRHS of the elderly. Based on Table 4, the well-being and SRHS of the elderly were interdependent, which suggests an endogenous effect between the two.

## 6. Discussion

Wu [81] studied the well-being and coping methods of elderly residents living in old apartments in Nanchong, Sichuan and Jinan, Shandong. They found no significant gender differences in any factors related to well-being. Jiang and Lin [82] surveyed the coping methods and factors influencing the subjective well-being of elderly people in Fuzhou, China. Similarly to our findings, the results showed no significant differences in the well-being perceived by elderly people in different age groups. An et al. [59] reported that people with a higher level of education often had a higher level of life satisfaction and well-being; well-educated people are more capable of managing their life problems. Huang and Yang [22] found that university graduates experienced a higher level of well-being than those who only graduated from elementary school or junior, senior, or vocational high schools. A possible reason for this is that university graduates have greater self-knowledge and can enrich themselves at any time and control their life. Our results do not support these findings. This shows that, as expected, married people had a higher level of well-being than those who were divorced, separated, widowed, or unmarried. Tan et al. [52] noted that compared to married or cohabitating people, separated or divorced people had significantly lower well-being. Yang and Leone [60] supported the argument that married people experience a better sense of well-being than unmarried people, possibly because marriage provides security, and people have a better sense of well-being as a result.

Hwang and Sim [30] found that elderly people who lived with their spouses reported higher levels of well-being compared to those living with their family or alone. Our results do not support those findings. Living with family members was associated with lower well-being, and the sign was not in line with expectations. One reason for this is that most elderly people expect to live with their children so that they can take care of one another. However, friction often occurs between family members living together because of different lifestyles and opinions, and this reduces well-being. (This description may not be generalized to other countries or regions due to sociocultural differences and differing values.) However, our empirical results in this regard were not statistically significant. Aykan and Wolf [83] found that for married adult children, co-living with their parents was not a fixed norm. Continuous economic developments and concomitant social changes have decreased the popularity of co-residence among parents and their adult children. Even though parents wish to live with their children so that they can care for one another, intergenerational family members who live together may often be at odds with one another because of their different lifestyles and values, thus decreasing well-being.

Cuong [61] demonstrated that having a higher income was conducive to elderly people’s satisfaction with life and other people. However, our empirical results do not support the argument that a higher level of well-being is influenced by income. Wu [84] empirically studied the relationship between income and subjective well-being, The factors influencing the latter include health and longevity, work and income, social relations, and social interest. The factors affecting subjective well-being were mainly evaluated based on the relative value of income instead of the absolute value. Therefore, if salary level remains the same, the perceived well-being differs based on the changes in the user’s residence and location.

An et al. [59] found that elderly people with high activity levels often experienced higher life satisfaction and well-being. Badri et al. [85] reported that SRHS could, for the most part, explain well-being. Our empirical results are consistent with the findings of [52,76] and in line with our expectations. A significant and positive relationship exists between well-being and SRHS; when SRHS is higher, well-being is greater, and elderly people with better well-being are healthier.

This shows that elderly people with sole or joint ownership of real estate had higher levels of well-being than those who did not own their homes, and those with sole ownership had the highest level of well-being. The findings are in line with expectations. Hu and Ye [86] reported that house ownership was positively correlated with a person’s aggregate well-being, and joint ownership with a spouse was associated with a higher level of well-being compared to those not owning their home or where the spouse had ownership. Chang [87] found that real estate ownership had a positive effect on SWB.

Okulicz-Kozaryn [88] examined US social survey data and found that millennials living in major cities and elderly people living in suburban areas had higher levels of well-being; millennials found Internet access and convenient transportation in the city more attractive. Tobiasz-Adamczyk and Zawisza [89] studied elderly Polish people living in rural and suburban areas and found that the SWB of villages significantly improved with the positive effects of social networks.

This suggests that elderly people who were aging in their own house or community had a higher level of well-being than those living in nursing homes, and those aging in the community had the highest well-being. Pozzi et al. [53] reported that elderly people could improve their sense of belonging in the community by participating in community activities; this social capital motivates them to stay in the community and improves their well-being. Zhang [35] studied the living arrangements and SWB of elderly Chinese people and revealed that living alone or in institutions was strongly associated with negative well-being. Yu [69] found that, in general, the well-being of those living in their community over time reached a moderate or high level. Elderly people in the study emphasized their quality of life, and more community-dwelling elderly are willing to step out of their homes and make friends, thus enriching their lives. Our empirical results are consistent with the findings of [29,46,62] and in line with our expectations.

Elderly people with high levels of activity participation were 4.512 times more likely to have high well-being than those with low levels of activity participation. Hwang and Sim [30] showed that participating in leisure activities was positively correlated with the well-being of the elderly. Yu [69] recommended that elderly people should actively participate in community activities because they can build their social support networks and enhance their SWB. Our results support these research findings.

Elderly people with high social support were 4.026 times more likely to have higher well-being than those with low social support. Patricia [90] revealed that providing and receiving support is conducive to the well-being of the elderly, especially when providing support, because they perceive themselves as in need. Glass [91] found that in light of the nursing shortage as well as alternative solutions to institutional care, mutual support is considerably important to community-dwelling elderly. Our results are in line with those of previous studies on the well-being of the elderly: high social support enhances their well-being [92,93,94].

## 7. Conclusions and Recommendations

In response to the rapid aging of its population, Taiwan’s Long-term Care 2.0 Plan focuses on aging in place and covers the choice of living arrangements by the elderly. In general, well-being entails interpersonal relations (or social capital). Few previous studies have examined the effects of living arrangements and social capital on the well-being of the elderly. The purpose of this study was to investigate the effects of aging in place and social capital on the well-being of the elderly. The relationships between well-being and the variables considered were analyzed using an ordered logit regression model, and the endogeneity arising from SRHS on well-being was resolved. While previous studies have examined the choice of living arrangements, the underlying issue of well-being remains overlooked. The relationship between living arrangements and well-being in elderly people can be analyzed more thoroughly using these three types of living arrangements (aging at home, aging in the community, and aging in a nursing facility).

### 7.1. Policy Implications

Previous studies [20,23] are more inclined toward the analysis of SWB through individual socioeconomic background. Our results suggest, however, that compared to personal attributes, social capital may be a more important factor explaining differences in SWB. The theoretical implications of SWB were examined in this study through social capital. The choice of living arrangements is impacted by the available resources and interpersonal networks, which is also the distinctiveness and contribution of this study with respect to the development of inter-variable relations. Previous studies on living arrangements mostly examined the means to implement the choice of aging in place and provide strategies for successfully doing so [95,96]. This study examines the impact of current living arrangements on the well-being of the elderly to compensate for the challenges to well-being when aging in place. Another feature of this study is the inclusion of endogeneity. From the perspective of economic theory, a causal relationship may exist between the dependent and independent variables, and there is thus a need to take into account the endogeneity of the well-being and SRHS of the elderly.

First, regarding the choice of living arrangements, a person is exposed to different types of people based on their location for aging. Our results demonstrated that elderly people who are aging in the community had the highest level of well-being, followed by those aging at home and receiving institutional care. As aging in place is advocated in today’s society, elderly people should interact with more people instead of staying at home; this will prolong the time they can live at home or in the community, allowing them to remain familiar with their neighborhood and have a more rigorous and autonomous lifestyle.

Furthermore, social support increases the well-being of the elderly. Social support improvement begins with participating in community and volunteer activities, attending a community college or a lifelong learning center, or connecting with friends and relatives. When participating in activities and interacting with others, elderly people feel content and less isolated, as they perceive a close interpersonal relationship with others.

Third, with regard to real estate ownership, sole ownership had a significant and positive impact on SWB. However, in practical terms, the concept that “along with land comes wealth” is prevalent in Asia, and social classes may exist based on real estate ownership. Owning real estate is conducive to a better quality of life and mental health; compared to tenants or those with no ownership, those who own real estate are subjected to less psychological stress, such as that arising from living in their landlord’s property.

Fourth, the variables SRHS and well-being are interdependent. It is recommended that elderly people maintain their physical fitness—frequent exercise is conducive to well-being. Having a healthy physique motivates one to achieve goals, and being mentally happy and content results in more favorable SRHS.

### 7.2. Recommendations for Future Research

This study collected data through a questionnaire at the height of the COVID-19 pandemic. In the future, a complete and comprehensive database could be established by administering questionnaires to elderly people who are receiving institutional care. In modern society, the Internet also entails social capital, and hence, the well-being of middle-aged and elderly people could be assessed in terms of the frequency with which elderly people go online [97,98]. The concept of social exclusion could also be used, as studies have shown that neighborhood exclusion is particularly significant for the elderly since it is associated with a poor quality of life [99]. Future studies could examine the impact of social exclusion on the well-being of the elderly.

## Figures and Tables

**Table 1 healthcare-11-02050-t001:** Description of variables.

Variable	Description	Expected Sign
Dependent variables		
Subjective well-being (SWB)	All items are measured on a five-point scale, with a higher score indicating a higher level of well-being. There are three categories: SWB = 1 (≤2.5 points); SWB = 2 (2.6~3.5 points); SWB = 3 (≥3.5 points).	
Self-rated health status (SRHS)	Current health status is measured through ADLs and IADLs. All items are measured on a five-point scale, with a lower score indicating poorer health. There are three categories: SRHS = 1 (≤2.5 points); SRHS = 2 (2.6~3.5 points); SRHS = 3 (≥3.5 points).	
Independent variables		
Personal attributes		
Sex (SEX)	Sex was set as a dummy variable (men = 1, women = 0).	+/−
Age (AGE)	Age was set as a dummy variable (65 to 75 years = 1, 76 years and above = 0).	+
Education level (EDU)	Education level is either low (elementary school and below), medium (junior high school and senior (vocational) high school), or high (university or postgraduate education), with a low level of education set as the baseline. There are two dummy variables: EDU1, for high education level = 1, and 0 otherwise; EDU2, for medium education level = 1, and 0 otherwise.	+
Marital status (MARRY)	Marital status is either married, widowed, divorced or separated, or unmarried. Marital status was set as a dummy variable (MARRY), with unmarried (widowed, divorced, or separated) as the baseline for comparison. Married = 1, and 0 otherwise (separated/divorced/widowed). The coefficient of the dummy variable of marital status (MARRY) is expected to be positive.	+
Family type (FAMTYPE)	Family type options are living with a spouse, living with family, or living alone, with living alone as the baseline for comparison. There are two dummy variables: FAMTYPE1, for living with spouse = 1, and 0 otherwise; FAMTYPE2, for living with family = 1, and 0 otherwise.	+
Personal monthly income (INCOME)	Personal monthly income is either low (less than TWD 22,000), middle (TWD 22,001 to TWD 38,000), or high (more than TWD 38,001) income, with low income as the baseline for comparison. There are two dummy variables (INCOME 1, for high income = 1, and 0 otherwise; INCOME 2, for middle income = 1, and 0 otherwise).	+
Exercise frequency (SPORT)	Exercise frequency is measured as once weekly or less, and twice, thrice, four times, and five times or more weekly. In the dummy variable of SPORT, three or more times weekly was set as 1, and 0 otherwise.	+
Habits	Habits considered are smoking, alcohol consumption, and betel nut chewing, with no habits being the baseline for comparison. Smoking is set as a dummy variable (SMOKE), for smokers = 1, and 0 otherwise. Alcohol consumption is set as a dummy variable (ALCOHOL); for drinkers = 1, and 0 otherwise. Betel nut chewing is set as a dummy variable (BETEL); for betel nut chewers = 1, and 0 otherwise.	+
Housing attributes:		
House ownership (COWNSHIP)	Real estate ownership is either sole ownership, joint ownership with a spouse or with parents, siblings, or children, and rental or spousal ownership, with rental or spousal ownership as the baseline for comparison. There were two dummy variables (COWNSHIP 1, for sole ownership = 1, and 0 otherwise; COWNSHIP 2, for joint ownership with spouse or with parents, siblings, or children = 1, and 0 otherwise).	+
Accessibility of medical facilities in living area (MEDICAL)	In this study, accessibility of medical facilities to the living area is set as a dummy variable (MEDICAL); where this is satisfied (3 points and above), this is set as 1, and 0 otherwise.	+
Living location (LOCATION)	Living location is set as a dummy variable (LOCATION), with living in suburban areas as the baseline for comparison. Living in the city center is set as 1, while living in suburban areas is set as 0.	+
Living arrangements (LIVETYPE)	Living arrangements include aging at home, aging in the community, or aging in a nursing facility, with living in a nursing facility as the baseline for comparison. Two dummy variables were set (LIVETYPE 1, for aging at home = 1, and 0 otherwise; LIVETYPE 2, for aging in the community = 1, and 0 otherwise).	+
Social capital: Social activities (SACTION)	The social activities in which the elderly participated included volunteering, leisure, learning, and religious activities. This participation was scored as one of four levels ranging from 1 to 4, indicating no participation, annual, monthly, and weekly participation, respectively. Based on the mean score, social participation was rated as either high or low, with low participation as the baseline for comparison. One dummy variable was set (SACTION) that for high participation = 1 and for low participation = 0.	+
Social support (SUPPORT)	Social support consists of no contact, occasional contact, and frequent contact, with no contact as the baseline for comparison. Two dummy variables were set (SUPPORT1, for frequent contact = 1, and 0 otherwise; SUPPORT2, for occasional contact = 1, and 0 otherwise).	+

**Table 2 healthcare-11-02050-t002:** Descriptive statistics of the sample (*n* = 362).

	Aging in Place (*n* = 213)		Community Elderly (*n* = 95)		Institutionalized Elderly (*n* = 54)		Total Percentage
	Frequency	Percentage	Frequency	Percentage	Frequency	Percentage	
Sex							
Male	119	62.96%	41	21.69%	29	15.34%	52.21%
Female	94	54.34%	54	31.21%	25	14.45%	47.79%
Age							
65–75 years	143	61.64%	49	21.12%	40	17.24%	64.09%
76–95 years	70	53.85%	46	35.38%	14	10.77%	35.91%
Education level							
No higher than elementary school	36	56.25%	22	34.38%	6	9.38%	17.68%
Elementary school	62	60.78%	30	29.41%	10	9.80%	28.18%
Junior high school	37	56.92%	19	29.23%	9	13.85%	17.96%
High school (vocational)	39	63.93%	9	14.75%	13	21.31%	16.85%
College	33	57.89%	10	17.54%	14	24.56%	15.75%
University	3	30.00%	5	50.00%	2	20.00%	2.76%
Postgraduate	1	100.00%	0	0.00%	0	0.00%	0.28%
Missing responses	2	100.00%	0	0.00%	0	0.00%	0.55%
Marital status							
Married	142	62.01%	62	27.07%	25	10.92%	63.26%
Widowed	56	52.83%	31	29.25%	19	17.92%	29.28%
Divorced or separated	14	70.00%	1	5.00%	5	25.00%	5.52%
Single	1	14.29%	1	14.29%	5	71.43%	1.93%
Family type							
Living alone	42	35.90%	21	17.95%	54	46.15%	32.32%
Living with spouse only	75	64.66%	41	35.34%	0	0.00%	32.04%
Living with family	95	74.80%	32	25.20%	0	0.00%	35.08%
Missing responses	1	50.00%	1	50.00%	0	0.00%	0.55%
Monthly household income (unit: TWD)							
<15,000	67	51.94%	40	31.01%	22	17.05%	35.64%
15,001–22,000	40	59.70%	18	26.87%	9	13.43%	18.51%
22,001–29,000	36	64.29%	10	17.86%	10	17.86%	15.47%
29,001–38,000	29	64.44%	9	20.00%	7	15.56%	12.43%
38,001–43,000	18	66.67%	7	25.93%	2	7.41%	7.46%
>43,001	22	66.67%	7	21.21%	4	12.12%	9.12%
Missing responses	1	20.00%	4	80.00%	0	0.00%	1.38%
Exercise frequency (weekly)							
Once or less	20	42.55%	17	36.17%	10	21.28%	12.98%
Twice	45	60.00%	9	12.00%	21	28.00%	20.72%
Thrice	34	53.13%	15	23.44%	15	23.44%	17.68%
Four times	33	66.00%	15	30.00%	2	4.00%	13.81%
Five times and above	81	64.29%	39	30.95%	6	4.76%	34.81%
Habits (multiple choice)							
Smoking	25	64.10%	4	10.26%	10	25.64%	10.77%
Drinking	18	52.94%	7	20.59%	9	26.47%	9.39%
Betel nut chewing	2	100.00%	0	0.00%	0	0.00%	0.55%
None	180	58.82%	84	27.45%	42	13.73%	84.53%
Housing ownership							
Rental	34	50.75%	15	22.39%	18	26.87%	18.51%
Sole ownership	86	58.50%	38	25.85%	23	15.65%	40.61%
Spouse’s ownership	42	64.62%	21	32.31%	2	3.08%	17.96%
Co-ownership with spouse	6	42.86%	5	35.71%	3	21.43%	3.87%
Co-ownership with family members other than spouse	45	70.31%	13	20.31%	6	9.38%	17.68%
Missing responses	0	0.00%	3	60.00%	2	40.00%	1.38%

**Table 3 healthcare-11-02050-t003:** Regression model fitting test.

		Predicted Response Category			Total
		1.00	2.00	3.00	
Subjective well-being (SWB)	1, low level of well-being	27 (55.1%)	22 (44.9%)	0 (0.0%)	49 (100%)
	2, medium level of well-being	9 (8.9%)	60 (59.4%)	32 (31.7%)	101 (100%)
	3, high level of well-being	0 (0.0%)	32 (15.1%)	180 (84.9%)	212 (100%)
Total		36 (9.9%)	114 (31.5%)	212 (58.6%)	362 (100%)
Self-rated health status (SRHS)	1, poor SRHS	7 (25.9%)	16 (59.3%)	4 (14.8%)	27 (100%)
	2, fair SRHS	4 (5.1%)	43 (54.4%)	32 (40.5%)	79 (100%)
	3, high SRHS	0 (0.0%)	16 (6.3%)	240 (93.8%)	256 (100%)
Total		11 (3.0%)	75 (20.7%)	276 (76.2%)	362 (100%)
		$\chi^{2}$		*p*-value	
		SWB	SRHS	SWB	SRHS
Test of Parallel Lines					
Null hypothesis					
Generalization		21.192	22.167	0.270	0.138
Model fitting		271.577	202.521	0.001	0.001

**Table 4 healthcare-11-02050-t004:** Empirical results.

	SWB Model			SRHS Model		
	β Coefficient	Wald Statistic	Odds Ratio (OR)	β Coefficient	Wald Statistic	Odds Ratio (OR)
Pre_SRHS	1.502 *** (0.146)	13.055	4.491			
Pre_SWB				0.957 *** (0.371)	6.643	2.604
SEX	−0.204 (0.310)	0.432	0.816	−00.383 (0.340)	1.273	0.682
AGE	−00.175 (0.394)	0.197	0.840	1.184 *** (0.333)	12.604	3.266
EDU1	−0.671 (0.492)	1.860	0.511	0.898 (0.587)	2.335	2.453
EDU2	0.187 (0.358)	0.273	1.205	0.555 (0.384)	2.095	1.743
MARRY1	0.933 ** (0.390)	5.716	2.541	−0.172 (0.446)	0.149	0.842
FAMTYPE1	0.048 (0.550)	0.007	1.049	0.188 (0.553)	0.115	1.206
FAMTYPE2	−00.129 (0.448)	0.082	0.879	−00.467 (0.388)	1.449	0.627
INCOME1	0.569 (0.612)	0.862	1.766	2.180 ** (1.109)	3.863	8.850
INCOME2	0.285 (0.357)	0.638	1.330	0.241 (0.392)	0.377	1.272
SPORT1				1.191 *** (0.388)	9.437	3.290
SMOKE				−00.168 (0.454)	0.137	0.846
ALCOHOL				−00.874 * (0.502)	3.034	0.417
BETEL				−024.396 (24,951)	0.000	2.541
COWNSHIP1	1.243 *** (0.346)	12.904	3.465			
COWNSHIP2	0.925 ** (0.364)	6.464	2.523			
MEDICAL				0.820 ** (0.400)	4.209	2.270
LOCATION	0.117 (0.278)	0.177	1.124	−0.778 ** (0.333)	5.450	0.459
LIVETYPE1	1.155 ** (0.536)	4.651	3.174			
LIVETYPE2	1.955 *** (0.587)	11.095	7.063			
SACTION	1.507 *** (0.482)	9.781	4.512			
SUPPORT1	1.393 ** (0.555)	6.292	4.026			
SUPPORT2	−0.082 (0.488)	0.028	0.921			

Note: * indicates a level of significance at *p* < 0.1; ** indicates a level of significance at *p* < 0.05; *** indicates a level of significance at *p* < 0.01. The number in brackets represents standard errors.

## Data Availability

The data presented in this study are available upon request from the corresponding author.

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
