# Peer review of "The Impact of Living Arrangements and Social Capital on the Well-Being of the Elderly"

_healthcare, 2023, doi:10.3390/healthcare11142050_

Round 1

Reviewer 1 Report (Previous Reviewer 1)

It is particularly important to pay attention to the happiness of the elderly in an increasingly aging society. Based on questionnaire survey data, the paper used an ordered logistic regression model to explore the impact of living arrangements and social capital on the well-being of the elderly. The data used in the paper is detailed and the selected method is appropriate. The viewpoint of the paper is clear, the argument is sufficient, and the conclusions drawn are also relatively reliable. 

Some details in the paper.

1. There are three Chinese characters in Table 3.

2.Table 3 in line 502 should be changed to Table 4.

3. Why did the paper choose Pingtung City and Kaohsiung City as case studies? The reasons need to be clarified.

Author Response

Reviewer 2 Report (Previous Reviewer 2)

Thanks again for the opportunity to review this paper and your thoughtful considerations of the points for review.  I feel that the paper has been enhanced by these.

I feel a minor proof read / edit will assist the paper greatly.  For example, where responses to the reviewers are included, these sections could be better linked to the text before an after in their introductory and concluding statements.  This is very minor, but will help give the argument momentum.

As stated above...

I feel a minor proof read / edit will assist the paper greatly.  For example, where responses to the reviewers are included, these sections could be better linked to the text before an after in their introductory and concluding statements.  This is very minor, but will help give the argument momentum.

Author Response

Reviewer 3 Report (Previous Reviewer 3)

I am very appreciated that authors have put many efforts on the revision, however, there are still some weakness in the conceptual framework and research design that have not been fully addressed:

1.Regarding the analytical framework developed by this study, authors argued that "Health status is an endogenous variable because it is affected by well-being." Although you mentioned that " health (SRHS) affects well-being, and well-being also affects health", the mutual effects of these two concpects have not been fully elaborated. And the comprehensive associations of independent variables with the two dependents have not been clearly presented.

2.Data analysis strategy is missing. This paper did not present the way of authors analyzing the questionnaire data. This should be supplemented. Furthermore, it is necessary to address the approach of exploring the mutual effects between health and well-being (the simultaneous model?) .

3. In terms of the sampling, how did authors decide the sample size and the way of sampling? What kind of sampling method did they use? Can the samples represent the total older population? The sampling issues could undermine the credibility of the results and conclusion.

4. Table 2 shows that there are still missing data in the dataset, then how did the authors deal with the missing data?

5. Regarding the format of writing, it is better to separate result and discussion, as the result section should focus on representing the statistical findings from the data analysis, while discussion section should focus on the explanation of the results.

no

Author Response

This manuscript is a resubmission of an earlier submission. The following is a list of the peer review reports and author responses from that submission.

Round 1

Reviewer 1 Report

Under the background of increasingly serious aging, such research topic is very meaningful and important. The econometric method used in this paper is very effective. The research conclusion well explains the reality, which is of great theoretical and practical value. The paper needs to be supplemented in the following aspects.

(1)Descriptive statistics of attribute data of all interviewees.

(2)The paper is slightly weak in explaining the impact of living arrangements and social capital on the well-being of the elderly. As the core part of the paper, this content should explain in detail how living arrangements and social capital affect the well-being of the elderly. It is better to have the support of interview cases.

Reviewer 2 Report

Thanks you for the opportunity to review this very interesting paper. I feel it makes a good contribution to debates around ageing in place and wellbeing.  There are a few discussion points below - mainly around how the argument is framed - but I feel that these only enhance the very interesting findings.

Lines 30-37: The literature here is quite broad, and the attitudes to both ageing and wellbeing varies a lot between nations and cultures. It would be good if this could be considered more, and in particular related to the culture your study is based in.

Lines 52-63: I think this paragraph could be enhanced by considering literature on a number of factors. Firstly, rather than the benefits of "living with their children", I think more focus should be placed on the benefits of intergenerational living in general. Secondly, I think considerations of the psychological relationship with concepts of home should be discussed. Finally, related to this, a discussion of unsuitable housing should be added.  The argument that people "should consider aging in a nursing facility or care centre" is problematic.  Why should they just "age" and not "live" somewhere? Why should they be forced to change, when they could be better supported by policy? This is a complex argument.

Line 94 - these points are well made, but need to be better related to ageing. We know that older populations are more likely to have multiple health issues, therefore exacerbating this issue.

Line 127 - a point about the stigma associated with institutional living should be added here.

Line 173 - I think betel nut chewing requires more justification for an international audience.

Line 254 - did no variable assess suitability of housing?  For example, those living with poor housing standards (hazards, overcrowding, etc.) have been shown to have considerably worse health / wellbeing.

Line 371 - some methodological reflection here would be helpful - why were these settings chosen, and why the use of in-person questionnaires?  How were people identified in parks, etc.?

Line 433 - be careful of the phrasing - some of these statements may be true in the context of the study, but are not universal (this is not an expectation in the UK, for example).

Reviewer 3 Report

The research topic is interesting and significant for achieving an age-friendly society. However, there are several concerns in terms of the scientific of this study: 

1. The theoretical review and conceptual framework should be improved , eg. is there any interaction between social capital and living arrangement? what's the influential mechanism of personal attributes, social capital, and living arrangements affecting wellbeing and health, and what are the most influential factors? The analytical framework should be developed from your literature review, while for now, more effects are put on describing the existing findings, lacking of critical arguments to some extent. 

2. The consistency between theoretical foundation and variables selection could be improved, eg. why housing attributes are selected as a group of dependent variables? I didn't see the necessity from the literature review. 

3. Section 5 needs major revision, as the authors mixed the results and discussion together. The description of results seems ok, however, the discussion parts were relatively weak. Although authors cited quite a lot extant studies to compare with their results, they failed to explain the underlying reasons for the certain results. For example, line 418, 431,443-444, 450, etc., they only argued that the empirical results are consistent or inconsistent with previous studies, however, they didn't give further explanation why. That's the major flaws of this study.   

Round 2

Reviewer 3 Report

Although the authors made some improvements on the manuscript, however, there are several key issues that need to be addressed before it can be considered for publication.

Firstly, the theoretical review is not satisfied. The research lacks depth in its conceptual framework, making it difficult to justify the rationality of the empirical analysis. Additionally, the theoretical framework needs to be much clearer to provide readers with a better understanding of the methodology.

Secondly, the analysis methods could be improved. There are numerous descriptive analysis with the data, while the regression analyses are lack of theoretical basis, making it difficult to draw meaningful conclusions from the study. In addition, authors addressed the statistical findings and discussion together, which is highly not recommended. 

Thirdly, the writing of literature review and discussion needs improvement. The texts tell in a simple, straightforward way, lacking of summary of the related standpoints and critical comments. So it is difficult for readers to grasp the main idea and follow author's arguments.

In summary, while the research presents some interesting ideas, the issues outlined above make it unsuitable for publication in its current state. I encourage authors to revise and resubmit your article after addressing these concerns.

Author Response

please see the attachement.
